# Group V Chitin Deacetylases Influence the Structure and Composition of the Midgut of Beet Armyworm, *Spodoptera exigua*

**DOI:** 10.3390/ijms24043076

**Published:** 2023-02-04

**Authors:** Han Wu, Dan Zhao, Xiao-Chang Guo, Zhao-Rui Liu, Rui-Jun Li, Xiu-Jun Lu, Wei Guo

**Affiliations:** 1College of Plant Protection, Hebei Agricultural University, Baoding 071001, China; 2Graduate School of Chinese Academy of Agricultural Sciences, Beijing 100081, China

**Keywords:** chitin deacetylase, *Spodoptera exigua*, RNA interference, midgut, peritrophic membrane

## Abstract

Chitin deacetylase (CDA) can accelerate the conversion of chitin to chitosan, influencing the mechanical properties and permeability of the cuticle structures and the peritrophic membrane (PM) in insects. Putative Group V CDAs SeCDA6/7/8/9 (SeCDAs) were identified and characterized from beet armyworm *Spodoptera exigua* larvae. The cDNAs of *SeCDAs* contained open reading frames of 1164 bp, 1137 bp, 1158 bp and 1152 bp, respectively. The deduced protein sequences showed that SeCDAs are synthesized as preproteins of 387, 378, 385 and 383 amino acid residues, respectively. It was revealed via spatiotemporal expression analysis that *SeCDAs* were more abundant in the anterior region of the midgut. The *SeCDAs* were down-regulated after treatment with 20-hydroxyecdysone (20E). After treatment with a juvenile hormone analog (JHA), the expression of *SeCDA6* and *SeCDA8* was down-regulated; in contrast, the expression of *SeCDA7* and *SeCDA9* was up-regulated. After silencing *SeCDAV* (the conserved sequences of Group V CDAs) via RNA interference (RNAi), the layer of intestinal wall cells in the midgut became more compact and more evenly distributed. The vesicles in the midgut were small and more fragmented or disappeared after *SeCDAs* were silenced. Additionally, the PM structure was scarce, and the chitin microfilament structure was loose and chaotic. It was indicated in all of the above results that Group V CDAs are essential for the growth and structuring of the intestinal wall cell layer in the midgut of *S. exigua*. Additionally, the midgut tissue and the PM structure and composition were affected by Group V CDAs.

## 1. Introduction

*Spodoptera exigua* (Hübner) is a moth (Lepidoptera) of the Noctuidae family and a considerable polyphagous economic pest in the world [1,2], causing damage to a variety of crops. *S. exigua* is found in all provinces, regions and cities in China. Seventeen Chinese provinces have reported *S. exigua* outbreaks, resulting in significant losses in agricultural production [3,4]. Chemical pesticides are currently the primary means of controlling the reproduction of *S. exigua*. However, because of an over-reliance on chemical pesticides, *S. exigua* resistance has grown fast. In some regions, a high level of cross-resistance to various pesticides has been developed in field populations in recent years [5,6,7].

The hydrolysis of chitin N-acetylamino linkages is catalyzed via chitin deacetylase (CDA, EC3.5.1.41), a chitin hydrolase that results in the formation of chitosan [8]. CDA is an extracellular chitin modification enzyme that deacetylates *β*-(1,4)-linked N-acetylglucosamine homopolymer chitin to produce *β*-(1,4)-linked D-glucosamine residue polymer chitosan [9]. Chitin is hydrolyzed by chitinase and N-acetyl-β-D-glucosaminidase, whereas CDA enzymatically alter chitin through a deacetylating process [10]. Many CDAs have been isolated from fungi, bacteria, viruses, and insects.

Insect CDA proteins were classified into five homologous groups (Groups I–V) based on phylogeny and CDA-conserved motifs [9]. Group I CDA proteins include CDA1 and CDA2, while Group II CDA proteins only contain CDA3. Group I and II CDA proteins have three domains: the chitin-binding functional domain (ChBD), the low-density lipoprotein-receptor-binding domain (LDLa), and the chitin deacetylation catalytic domain (CDA). The CDA4 protein of Group III and the CDA5 protein of Group IV have ChBD and chitin deacetylation catalytic domains, with an extended intermediate region separating the ChBD and chitin deacetylation catalytic domains. The CDA6/7/8/9 protein of Group V only has the chitin-deacetylated catalytic domain [9]. Five conserved catalytic motifs exist in the chitin deacetylation domain of CE-4 family proteins in the insect catalytic domain [11]. During the development of insects, different CDA groups serve various purposes according to tissue and developmental stages. All nine *CDAs* have been found and characterized in *Tribolium castaneum* [12]. The knockdown of *TcCDA1* or *TcCDA2* via RNA interference (RNAi) caused a considerable molting failure phenomenon. However, after the knockdown of TcCDA3 to TcCDA9, no developmental problems were detected [12]. Four CDA genes were also discovered in *Nilaparvata lugens* using whole-genome sequencing [11]. Double-stranded RNA (dsRNA) injection for *NlCDA3* resulted in no discernible morphological abnormalities, while molting failure and significant mortality were caused by the silencing of *NlCDA1*, *NlCDA2* and *NlCDA4* [11].

CDAs appear widespread in insects and have been found in various taxa. Guo et al. discovered an insect CDA protein as a novel peritrophic membrane (PM) protein, TnPM-P42, from the cabbage looper *Trichoplusia ni*, which exhibits chitin-binding activity [13]. Since then, many insect species from different groups, including Lepidoptera [10,14,15,16,17,18], Coleoptera [12,19,20,21], Orthoptera [22], Hemiptera [11], Hymenoptera [9], Diptera [23,24] and Isoptera [25], have been shown to contain CDA genes. Several CDAs in *Bombyx mori* and *Mamestra configurata* are primarily expressed in the PM and play significant roles in changing the PM’s physical characteristics [26,27]. The CDA protein in the PM of *Helicoverpa armigera* probably protects the gut from external infection and intercepts toxins, such as *Bacillus thuringiensis* (Bt) crystal proteins [28]. The silkworm midgut genes *BmCDA6*, *BmCDA7* and *BmCDA8* exhibit unique expressions in the anterior midgut. However, their temporal expression pattern is different [29]. The function of Group V CDAs and their role in the structure of the insect midgut was speculated upon in previous studies, but not in depth [26,27,28,29]. The functions of Group V CDAs SeCDA6/7/8/9 (SeCDAs) have not been clarified in *S. exigua*.

In this study, the genome database of *S. exigua* was used to identify and clone *SeCDAs*. The predicted amino acid sequences, protein domains and phylogenies of the genes were then investigated. The expression profiles of the *SeCDAs* in tissue and developmental expression were investigated using qRT-PCR. RNAi technology was used to silence *SeCDAs* to explore their biological role in *S. exigua* and to elucidate the mechanism of CDA action. These findings will facilitate our understanding of the functions of Group V CDAs and serve as the foundation for further research.

## 2. Results

### 2.1. Group V SeCDAs Sequence Analysis

Nucleotide sequences of the cDNAs for Group V SeCDAs were obtained from the *S. exigua* genome (Table 1). It was shown by the NCBI Blastp comparisons that SeCDAs only have a CDA domain (Figure 1). It was demonstrated by the deduced protein sequences that SeCDAs proteins are synthesized as preproteins with signal peptides using the SignalP 5.0 Server. The secreted SeCDAs contain multiple *N*-glycosylation sites and five conserved motifs of the CDA domain (Appendix A).

A phylogenetic analysis was performed based on the CDA amino acid residue sequences of several insect species and the SeCDAs amino acid sequence of *S. exigua* published in NCBI GenBank (Figure 2, Appendix A). It was shown by the analysis that SeCDAs cluster among Group V CDA proteins. There was minimal genetic distance to the Lepidoptera species of *Spodoptera litura, Spodoptera frugiperda* and *Helicoverpa armigera*.

### 2.2. Group V SeCDAs Expression Analyses

It was shown via qRT-PCR analysis that Group V *SeCDAs* are expressed in different developmental stages of *S. exigua*. *SeCDA6* gene expression was highest in eggs, followed by the fifth- and second-instar larvae. Expression of *SeCDA7* and *SeCDA9* genes was found in the fifth-instar larvae. The *SeCDA8* gene was highly expressed in eggs and prepupae (Figure 3).

The expression of *SeCDAs* in different tissues of the fifth-instar day-2 larvae was shown via qRT-PCR analysis. *SeCDA*s were expressed in all tissues of the fifth-instar larvae and were highly expressed in the midgut (Figure 4A). The *SeCDAs* expression was more abundant in the anterior region of the midgut, followed by the middle and posterior regions of the midgut (Figure 4B).

### 2.3. Expression of Group V SeCDAs in Response to 20E and JHA Signaling

Furthermore, qRT-PCR was used to analyze the effects of the injection of 20-hydroxyecdysone (20E) and juvenile hormone analog (JHA) on the *SeCDAs* of *S. exigua*. The injection resulted in the down-regulation of the transcriptional expression level of *SeCDAs* when treated with 20E (Figure 5A). The expression of *SeCDA6* and *SeCDA8* decreased after JHA treatment. However, the expression of *SeCDA7* and *SeCDA9* increased after JHA treatment (Figure 5B).

### 2.4. Functional Analysis of Group V SeCDAs

qRT-PCR analysis showed that the injection of ds*SeCDAs* can cause the silencing efficiency of *SeCDAs* to be 79.28%, 75.55%, 73.23% and 87.57%, respectively. The relative expression level of *SeCDAs* was reduced by 99.89%, 90.45%, 84.46% and 96.60% after the injection of ds*SeCDAV* (Figure 6). Group V *SeCDAs* were silenced in larvae injected with ds*SeCDA6* and ds*SeCDA8*, respectively. *SeCDA6* and *SeCDA7* were silenced when ds*SeCDA7* was injected into the fourth-instar larvae. Only the *SeCDA9* gene was silenced when ds*SeCDA9* was injected. This was indicated by the results that the silencing efficiency of *SeCDAs* can be up-regulated via ds*SeCDAV* injection. After the *GFP* and *SeCDAs* were silenced, the larvae were molted and pupated normally (Appendix A).

### 2.5. Immunohistochemistry

The specific signal of SeCDAs was weak in the midgut after ds*SeCDAV* injection, which confirmed the specificity and silencing effect of the SeCDAs antibody. The immunohistochemistry method was used to explore the biological role of SeCDAs. Group V SeCDA protein was localized on the intestinal wall cells of the midgut of *S. exigua*. After silencing *SeCDAs*, the intestinal cell layer became thinner, and the intestinal cells were more closely arranged (Figure 7).

### 2.6. Scanning Electron Microscopy Analysis

Seventy-two hours after the injection of ds*SeCDAV*, the morphology of the midgut and PM were observed under a scanning electron microscope. The results showed that the vesicles on the midgut epithelia disappeared after the injection of ds*SeCDAV* compared with the control injected with ds*GFP* (Figure 8A). The PM treated with ds*SeCDAV* became rough with large folds and pores; in contrast, the chitin microfilament structure was loose and disordered (Figure 8B).

## 3. Discussion

According to the GenBank databases, five groups of CDA proteins have been classified according to the alignment of their predicted chitin deacetylase domain sequences. SeCDA6/7/8/9, which only contains a CDA catalytic domain, belongs to the Group V CDAs protein (Figure 1).

It was indicated by recent studies on CDA in insects that the abundance of these proteins was different in the tissue, developmental stages and species. The first CDA protein, TnPM-P42, a midgut-specific protein, was identified from *T. ni* in 2005 [13]. In subsequent studies, Group I CDA was mainly found in ectodermal tissues, such as the foregut, hindgut, integument and trachea. Group III CDA occurred primarily in specialized epidermal tissues, and Group V CDAs were mainly expressed in the midgut-specific proteins. BmCDA6/7/8 were specifically described in the anterior part of the midgut depending on the larval instar. The expression of BmCDA6 was concentrated primarily in the middle-molting stage. BmCDA7 and BmCDA8 were only highly expressed in the feeding stages. [29]. TcCDA6/7/8/9 were expressed specifically in the midgut during the larval feeding stage [12]. *SeCDAs* identified from *S. exigua* were more abundant in the most anterior region of the midgut, which was similar to Group V CDAs in *B. mori* [29] and *T. castaneum* [12]. This indicated that SeCDAs were primarily synthesized in the anterior part of the midgut. We speculate that SeCDA7 and SeCDA9 may be involved in modifying and maintaining midgut characteristics to stabilize the metabolism. Moreover, SeCDA6 and SeCDA8 proteins were more frequently detected in the eggs. We speculate that SeCDA6 and SeCDA8 may be involved in midgut formation during egg development and the modification of their structures.

In insect, the bioactive hormone 20E and juvenile hormone (JH) play a crucial role in development, metamorphosis and reproduction. Insect molting may be induced by this process [30] and regulate specific gene expression [31,32,33], such as five chitin synthesis pathway genes in *S. exigua* [34]. Insect egg maturation is triggered by JH, which also helps maintain the larval condition, adult sexual development regulation, and premature larval metamorphosis prevention [35]. The JHA was used instead of JH in the larvae. The treatment of 20E accounted for a significant increase in mRNA expression levels of *BmCDA6*; however, it decreased the expression levels of *BmCDA7* and *BmCDA8*. The treatment of JHA up-regulated the expression of *BmCDA7* and *BmCDA8* [16]. Some results obtained after the treatment of 20E in other insects also regulated the *CDA* gene expression [16,36,37]. In this study, after the larvae of *S. exigua* were injected with 20E, the expression level of the *SeCDA6*/*7*/*8*/*9* was decreased. After treatment with JHA, the expression of *SeCDA6* and *SeCDA8* decreased, while *SeCDA7* and *SeCDA9* expressions increased. These results indicate that both 20E and JHA can regulate the expression of *CDA* genes.

RNAi technology has been widely used to regulate post-transcriptional genes in fungi, nematodes, and insects. According to their functions, insect CDAs can be divided into intestinal-specific proteins (Group V CDA) and parenteral-specific proteins (Groups I, II, III and IV CDA). The TcCDA6/7/8/9 were only expressed in the midgut, which belongs to midgut-specific proteins. There was no aberrant phenotype in individuals treated with ds*TcCDA6*/*7*/*8*/*9* [12]. RNAi was used to silence *Mythimna seperata CDA1* to evaluate PM alterations [38]. It was shown in the investigation that in the treated specimens, the dsRNA concentration was raised, the PM holes were wide, and the chitin microfilament structure became looser [38]. In our study, the silencing effectiveness of the *SeCDA6* gene following RNAi treatment was 79.28%. The *SeCDA7* gene silencing effectiveness was 75.55%. *SeCDA8* silencing efficacy was 73.23%, and *SeCDA9* gene silencing effectiveness was 87.57%. After the injection of ds*SeCDAV*, the silencing efficacy of *SeCDA6*/*7*/*8*/*9* was 99.89%, 90.45%, 84.46% and 96.60%, respectively. The silencing efficiency of all Group V *CDAs* was considerably better than that of any Group V *CDA* alone, and a new way to improve the silencing efficiency of Lepidoptera genes was provided. The expression of *SeCDAs* decreased in all larvae injected with ds*SeCDA6* or ds*SeCDA8*. Additionally, the DNA sequence alignment identity of *SeCDA6* and *SeCDA8* was as high as 93.64%.

This result may be due to the high similarity of the gene sequences. The silence of the *SeCDA7* gene led to decreased gene expression of *SeCDA6* and *SeCDA7*. Only the *SeCDA9* gene was silenced when ds*SeCDA9* was injected into the larvae. The DNA sequence alignment similarity between *SeCDA6* and *SeCDA7* was 53.95%. The sequence alignment similarities between *SeCDA9* and *SeCDA6*, *SeCDA7*, and *SeCDA8* are 65.64%, 60.88%, and 65.49%, respectively. The sequence similarity between *SeCDA7* and *SeCDA8* was only 55.53%. It was suggested by these results that the desired precise targeting effect may not be produced by highly similar gene sequences or the dsRNA targeting the same domain. The reasons for these results are unclear, and more research will be studied in the future.

*BmCDA8* was discovered in both the PM and the ectoperitrophic gap between the midgut epithelial cells and the PM. *BmCDA7* was discovered in the PM of *B. mori* [27,39,40]. These midgut CDAs may have been involved in PM production in the anterior midgut, where PM synthesis begins [41,42]. The structure of the chitin matrix in many insects is impacted by the deacetylation process [24,43,44]. It has been noted that PM permeability in vitro may be boosted by recombinant HaCDA5a, a CDA from the moth *H. armigera* [45]. In this study, we examined the localization of SeCDAs using immunohistology to determine their biological function. SeCDAs are present in the intestinal wall cells of the midgut of *S. exigua*. When *SeCDAs* were silenced, the intestinal cell layer shrank, and the intestinal cells were more evenly distributed. Group V CDAs were demonstrated by these findings to be critical in developing and organizing the intestinal wall cell layer in the midgut of *S. exigua*.

The maintenance of PM mechanical strength in *H. armigera* is probably correlated with CDA [17]. A change in the peritrophic membrane might make insects more prone to infection, which may eventually be lethal [46]. It was shown via scanning electron microscopy (SEM) analysis of the surface structure of the *S. exigua* midgut after ds*SeCDAV* treatment that the microvilli on the epithelia of the larval midgut did not alter substantially. However, all vesicles disappeared after the injection of ds*SeCDAV* compared to the control. This demonstrates that CDAs do not damage the microvilli but are crucial to the development of vesicles. Compared with the control, the PM of the larvae injected with ds*SeCDAV* became coarse, with many folds, pores and gaps. The chitin microfilament structure became loose and disorderly, similar to the results of *CDA* gene silencing in *M. separata* [38]. Therefore, it is hypothesized that CDA is crucial for maintaining the structural integrity of the PM of *S. exigua*. Group V *CDAs* are essential in forming the midgut and PM of *S. exigua*. However, no phenotypic changes were observed during molting in *S. exigua* with silenced Group V *CDAs*. The silencing of Group V *CDAs* affected the structure of the midgut and the PM. It was demonstrated by all the above results that the PM structure was disrupted when the Group V *CDAs* were silenced.

## 4. Materials and Methods

### 4.1. Insect Larvae

Larvae of *S. exigua* were reared on an artificial diet by the Laboratory of Insect Molecular Biology at Hebei Agricultural University. The rearing temperature was 26.5 ± 1 °C, the relative humidity from the egg to the third-instar larvae was 75 ± 10%, the relative humidity from the third-instar larva to the pupal stage was 65 ± 15%, and they were reared under a 14 L:10 D photoperiod. Fifth-instar larvae were used for dissection to isolate the PMs and various tissues for analyses.

### 4.2. Sequence Analysis

The sequence of the gene encoding Group V CDA was analyzed using Clustal X 1.83 and GeneDoc. The gene structure was mapped using the Exon-Intron Graphic Maker (http://www.wormweb.org/exonintron (accessed on 17 May 2021)). The open reading frame, coded amino acid, isoelectric point (pI) and molecular protein weight were predicted using the DNAMAN 6.0 program (LynnonBiosoft, USA). The NetNGlyc 1.0 Server (http://www.cbs.dtu.dk/services/NetNGlyc/ (accessed on 18 February 2022)) was used to predict the N-glycosylation sites. The TMHMM Server v.2.0 (https://services.healthtech.dtu.dk/service.php?TMHMM-2.0 (accessed on 19 February 2022)) was used to determine whether it belonged to a transmembrane protein, and SignalP 5.0 (https://services.healthtech.dtu.dk/service.php?SignalP-5.0 (accessed on 19 February 2022)) was used to predict the presence or absence of signal peptides. The NCBI Conserved Domain Database (CCD, https://www.ncbi.nlm.nih.gov/Structure/cdd/wrpsb.cgi (accessed on 23 June 2021)) was used to predict the conservative protein structure domain of SeCDA. Clustal X 1.83 was used for multi-sequence alignment, MEGA 5.0 software was used to build a neighbor-joining (NJ) phylogenetic tree, and 1000 repetitions were calculated using the bootstrap method, and the model was the Poisson model.

### 4.3. Gene Expression Analyses

Five tissues (integument, midgut, hindgut, malpighian tubules and fat body) were dissected from fifth-instar larvae for total RNA extraction to determine the tissue-specific expression pattern of *SeCDAs*. The hindgut and malpighian tubules from 10 larvae and other tissues (integument, midgut and fat body) from three larvae were pooled as one treatment. Three biological replications were prepared (each with three technical replicates). Eggs, first-, second-, third-, fourth-, and fifth-instar larvae, prepupae, and pupae were collected to determine their expression profiles during development. Each sample contained 3–20 individuals. Samples from each developmental stage were analyzed in biological triplicate. Each replicate was given a sample of at least 100 mg of tissue or larvae. The samples were stored at −80 until all samples were collected for RNA extraction.

Total RNA from the frozen tissues was isolated using the RNAprep Pure Tissue Kit (TIANGEN, Co., Ltd., Beijing, China) and treated with DNase I. A spectrophotometer (BioSpectrometer fluorescence, Eppendorf, Germany) and 1.2% agarose gel electrophoresis were used to measure RNA concentration and quality. The entire experimental operation was conducted in an RNase-free environment. A total of 1 μg of total RNA per reaction was used to perform cDNA synthesis using the PrimeScript RT Reagent Kit with gDNA Eraser (Perfect Real Time) (TaKaRa, Beijing, China) with an Oligo dT Primer.

The primers for qRT-PCR were designed using DNAMAN 6.0 software and are listed in Table 2. The melting curve was determined for each sample to detect the gene-specific peak and verify the absence of primer dimers. qRT-PCR was carried out in a 20 μL reaction volume containing 1 μL of cDNA, 10 μM of each primer, 7 μL of H_2_O and 10 μL of TB Green^®^ Premix Ex Taq™ II (Tli RNaseH Plus), ROX plus (TaKaRa, Beijing, China). PCR was conducted on a CFX-96 real-time PCR detection system (Bio-Rad, Hercules, CA, USA). The amplifications were performed using an Eppendorf thermocycler. They comprised an initial denaturation step at 95 °C for 3 min, followed by 39 cycles at 95 °C for 15 s, 55 °C for 30 s and 72 °C for 30 s, and a final extension step of 5 min at 72 °C. Relative transcript levels of target genes were calculated with the 2^−ΔΔCt^ method. Three biological repeats, each with three technical replicates, were set for each cDNA, and *β*-actin was used as the internal reference gene to calibrate the total amount of RNA. *β*-actin proved to be a suitable housekeeping gene applied in the qRT-PCR of *S. exigua* in our previous study [46]. The amplification efficiencies of the target and reference were approximately equal (Appendix A). The significance of differences between samples was assessed using SPSS 22.0 software (SPSS Inc., Chicago, IL, USA), and the data were presented as means and standard errors (SE).

### 4.4. Preparation and Treatment with 20E and JHA

A total of 40 fourth-instar larvae of *S. exigua* with similar body weights and body shapes were selected for the experiment. 20E (Solarbio, Beijing, China) was dissolved in dimethyl sulfoxide (DMSO) at a working concentration. A total of 5 µL of a 100 ng/µL 20E solution was injected into the fourth-instar larvae on the second day, and DMSO was injected into the control group. JHA (Sigma-Aldrich, Shanghai, China) was dissolved in dimethyl ketone (DMK). JHA and DMK were injected into *S. exigua* as described for the 20E treatment and control. RNA was extracted from whole *S. exigua* larvae after 12 h of treatment. Reverse transcription was performed, and the expression of *SeCDAs* genes was analyzed using qRT-PCR.

### 4.5. Functional Analysis of Group V CDA Genes

RNAi was used to reveal the biological roles of *SeCDA6*, *SeCDA7*, *SeCDA8* and *SeCDA9* in the midgut and PM of *S. exigua*. The primers containing the T7 RNA polymerase promoter sequences were designed with DNAMAN (Table 2). PCR was conducted to prepare the cDNA template for the synthesis of the dsRNA of *SeCDA6*, *SeCDA7*, *SeCDA8*, *SeCDA9*, *SeCDAV* (the conserved sequences of Group V CDAs, *SeCDA6*: 325 bp-813 bp, *SeCDA7*: 304 bp-786 bp, *SeCDA8*: 325 bp-807 bp, *SeCDA9*: 316 bp-798 bp) and *GFP* (control). The PCR product was verified on a 0.8% agarose gel. Samples (2 μg) were collected for ds*SeCDA6*, ds*SeCDA7*, ds*SeCDA8*, ds*SeCDA9*, ds*SeCDAV* and ds*GFP* synthesis using the T7 RiboMAX Express RNAi System (Promega, Madison, WI, USA).

A total of 150 fourth-instar 1-day larvae were collected and randomly divided into five groups. Aliquots of 10 μg of ds*SeCDA6*, ds*SeCDA7*, ds*SeCDA8*, ds*SeCDA9*, ds*SeCDAV* and ds*GFP* were injected into the hemocoel of the abdomen between the second and third segments with a microinjector. The silencing efficiency of ds*SeCDA6*, ds*SeCDA7*, ds*SeCDA8*, ds*SeCDA9* and ds*SeCDAV* was detected 72 h after dsRNA injection via qRT-PCR. Three biological replications (each with three technical replicates) were used to determine the silencing efficiency. The remaining larvae were maintained for phenotypical observation. The visible phenotypical changes were recorded daily until the larvae molted into adults.

### 4.6. Immunohistochemistry

Fourth-instar larvae were injected with ds*GFP* or ds*SeCDAV* separately to determine the specificity of the SeCDAs antibody. At 72 h after injection, the larvae were collected to prepare paraffin sections. Immunohistology was conducted as follows: *SeCDAs* were detected with the Group V SeCDAs rabbit antiserum (1:200) as primary antibodies and HRP Rabbit IgG/FITC (1:200) (Servicebio, Wuhan, China) as the secondary antibody for fluorescence detection. Propidium iodide (PI) is an analog of ethidium bromide that binds strongly to DNA and emits red fluorescence upon intercalation in double-stranded DNA, which stains the nucleus red. PI (Servicebio, Wuhan, China) was used for nuclear staining. Images were captured using a transmission electron microscope (HT7800/HT7700, Nikon, Japan) with excitation at 465–495 nm (HRP Rabbit IgG/FITC) and 330–380 nm (PI).

### 4.7. Scanning Electron Microscopy

The midgut and the PM of *S. exigua* larvae injected with ds*GFP* and ds*SeCDAV* were rinsed gently with PBS (Servicebio, Wuhan, China). The midgut and PM were quickly fixed using an electron microscope fixative (Servicebio, Wuhan, China) at room temperature for 2 h and transferred to 4° for storage. The midgut was washed, and PM was used three times for 15 min each in 0.1 M PBS. The tissues were placed in 1% OsO_4_ for 2 h at room temperature (Ted Pella Inc., Redding, CA, USA) in 0.1 M PBS. The tissues were washed three times for 15 min each in 0.1 M PBS. The tissues were sequentially added to 30%, 50%, 70%, 80%, 90%, 95% and 100% alcohol and isoamyl acetate for 15 min each time. The samples were dried with a Critical Point Dryer (K850, Quorum, UK) and sputter-coated with gold for 30 s with Lon Sputtering Apparatus (MC1000, Nikon, Japan). The tissues were observed under a scanning electron microscope (SU8100, Nikon, Japan), and the images were stored for analysis [38].

## 5. Conclusions

Group V SeCDAs (SeCDA6/7/8/9) were identified and characterized from the beet armyworm, *S. exigua*. The predicted proteins exhibited conserved structural features. *SeCDA*s were expressed at different developmental stages and in different larval tissues of *S. exigua*, and their expression was induced by injecting 20E or JHA. RNAi-mediated *SeCDAs* silencing considerably inhibited the expression of *SeCDAs* and caused structural changes in the midgut and PM. A reference for improving the silencing efficiency of Lepidoptera and enriching the biological function of Group V CDAs may be provided by the results of this study.

## Figures and Tables

**Figure 1 ijms-24-03076-f001:**
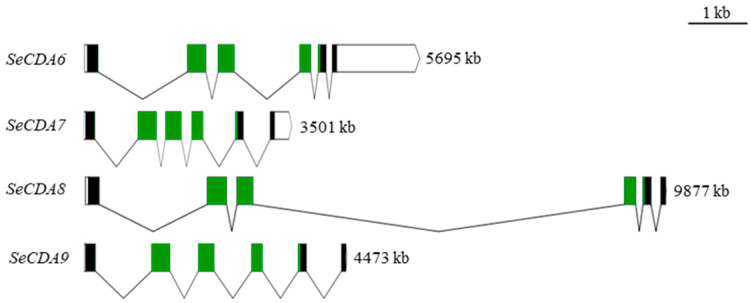
Schematic diagram of exon–intron of *SeCDAs* gene and domain of SeCDAs amino acid. The structure of the genome sequence is plotted to scale. Exons are represented by solid rectangles, noncoding regions are represented by hollow rectangles, and the connecting line between two exons represents introns. The green rectangles represent the chitin deacetylation catalytic domain.

**Figure 2 ijms-24-03076-f002:**
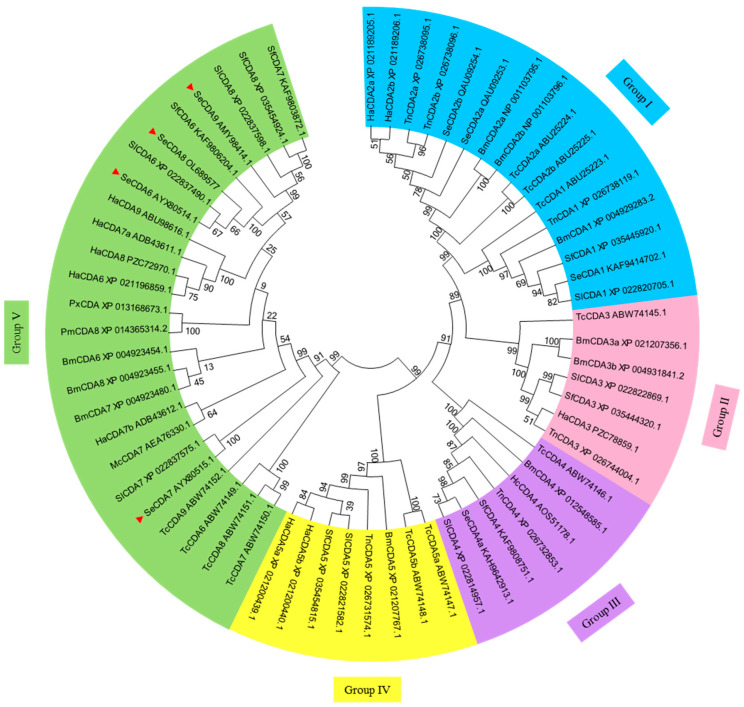
Phylogenetic tree of the CDA amino acid residue sequence of *S. exigua* and other insects. SeCDAs of *S. exigua* are labeled with a solid red triangle. The origin of CDA species: Bm: *Bombyx mori*; Ha: *Helicoverpa armigera*; Hc: *Hyphantria cunea*; Mc: *Mamestra configurata*; Pm: *Papilio machaon*; Px: *Papilio Xuthus*; Sf: *Spodoptera frugiperda*; Sl: *Spodoptera litura*; Tc: *Tribolium castaneum*; Tn: *Trichoplusia ni*.

**Figure 3 ijms-24-03076-f003:**
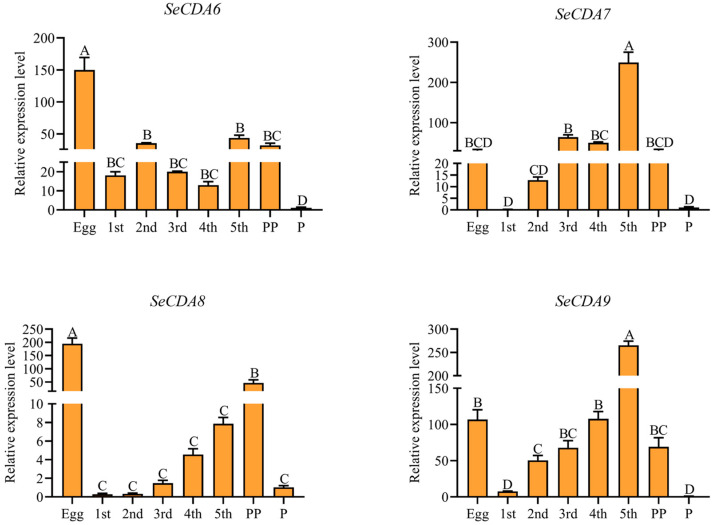
Expression profiles of *SeCDAs* in different developmental stages. Developmental stages include the egg, first-to-fifth-instar larvae, prepupae and pupae. Data were calibrated by using *β-actin* and presented as means ± SE of three separate experiments. The group of the pupae data was used for normalization. Different letters above the bars represent the different significances of the samples (*p* < 0.01).

**Figure 4 ijms-24-03076-f004:**
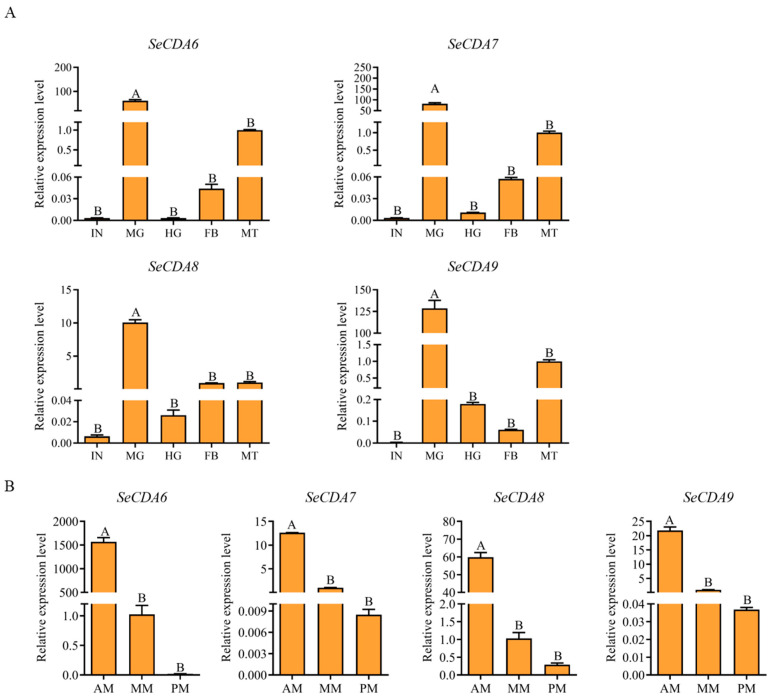
Detection of *SeCDAs* in various tissue samples and different midgut regions via qRT-PCR analysis. (**A**) IN: integument, MG: midgut, HG: hindgut, MT: malpighian tubule, FB: fat body. The MT data group was used for normalization. (**B**) AM: anterior part of the midgut, MM: middle part of the midgut, PM: posterior part of the midgut. The MM data group was used for normalization. Data were calibrated by using *β-actin* and presented as means ± SE of three separate experiments. Different letters above the bars represent the different significances of the samples (*p* < 0.01).

**Figure 5 ijms-24-03076-f005:**
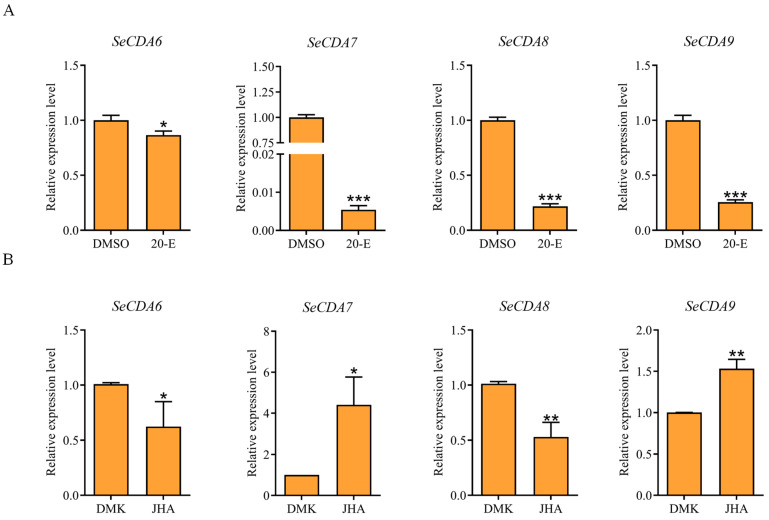
Effect of two hormone analogs on the expression of *SeCDAs*. (**A**) Effect of 20E on the expression of *SeCDAs*. DMSO was used in the control group. (**B**) Effect of JHA on the expression of *SeCDAs*. DMK was used in the control group. Data are reported as means ± SE of three independent biological replications (*** *p* < 0.001; ** *p* < 0.01; * *p* < 0.05).

**Figure 6 ijms-24-03076-f006:**
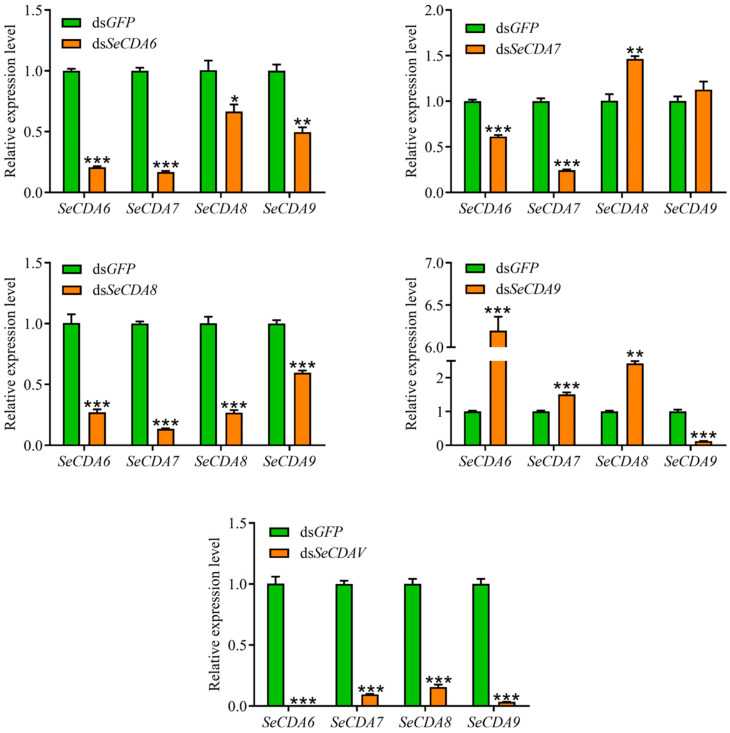
Relative expression levels of *SeCDAs* in *S. exigua* larvae injected with dsRNA-*SeCDAs*/*GFP*. Data are reported as means ± SE of three independent biological replications. (*** *p* < 0.001; ** *p* < 0.01; * *p* < 0.05).

**Figure 7 ijms-24-03076-f007:**
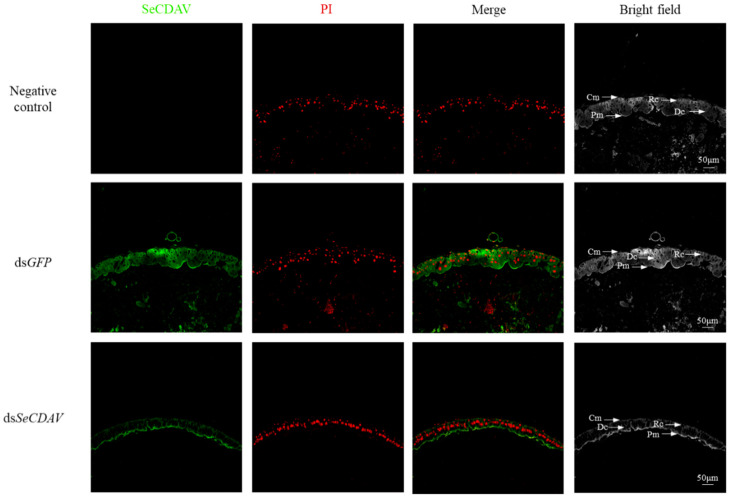
Detection of SeCDAs using a specific antibody. The specific signal of SeCDAs was detected after ds*GFP* and ds*SeCDAV* injection into fourth-instar larvae. The negative control was with the pre-immune serum. The SeCDAs protein is represented by green and the cell nucleus by red. Cm: circular muscle; Pm: peritrophic membrane; Rc: regenerative cells; Dc: digestive cells; scale bar is 50 μm.

**Figure 8 ijms-24-03076-f008:**
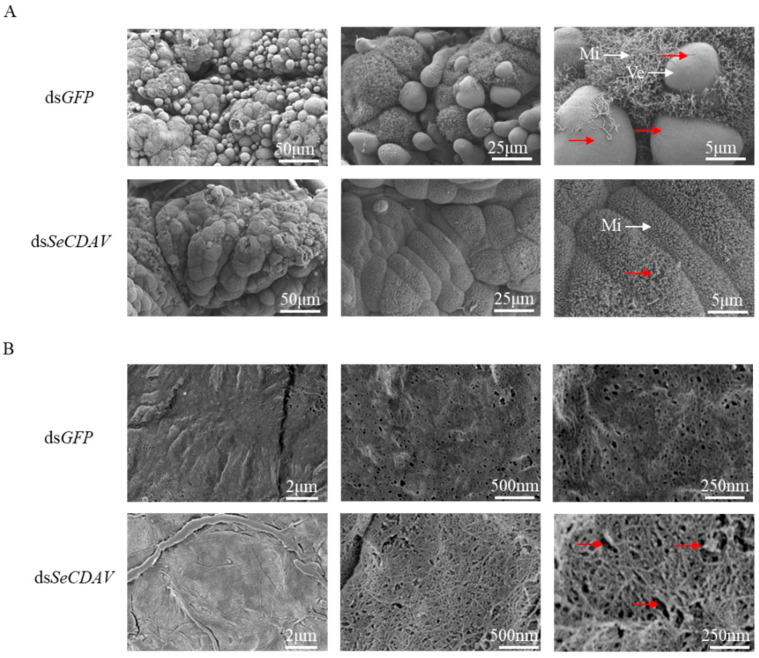
The ultrastructure of *S. exigua* larvae injected with dsRNA-*SeCDAV*/*GFP*. (**A**) Midgut ultrastructure of *S. exigua* larvae injected with dsRNA-*SeCDAV*/*GFP*. Red arrowheads point to vesicles. (**B**) Peripheral membrane ultrastructure of *S. exigua* larvae injected with dsRNA-*SeCDAV*/*GFP*. Red arrowheads point to the pores. Mi: microvilli; Ve: vesicles. The scale bars were 50 μm, 25 μm, 5 μm, 2 μm, 500 nm, and 250 nm.

**Table 1 ijms-24-03076-t001:** Summary of Group V *SeCDAs* in *S. exigua*.

cDNA	GenBank Accession	cDNA Length (bp)	ORF (aa)	M.V (kDa)	p*I*	Chromosome Position
SeCDA6	AYX80514.1	1164	387	44	4.82	Chr24
SeCDA7	AYX80515.1	1137	378	43	5.29	Chr24
SeCDA8	OL689577	1158	385	44	5.19	Chr24
SeCDA9	AMY98414.1	1152	383	43	4.98	Chr24

**Table 2 ijms-24-03076-t002:** Primers used in this study.

Primers	Sequences	Purpose
*SeActin*F	TTCCCATCCATCGTAGGT	qRT-PCR
*SeActinR*	GGATACCTCTCTTGCTCTGG	
*SeCDA6*F-Q	GAGTGCTTTGCTATTAATTCGC	
*SeCDA6*R-Q	GGTAGATAACGCCCTCAAGG	
*SeCDA7*F-Q	CGAAGAGTTGTGTAAGTTGCCT	
*SeCDA7*R-Q	CGAGTGTAAGGCGATTTCA	
*SeCDA8*F-Q	ATGTTACTCCTTGAAGCCGTTC	
*SeCDA8*R-Q	GGAACTGAAATTGCACTGGTG	
*SeCDA9*F-Q	CCTGGACTCTGGCCTTACACTC	
*SeCDA9*R-Q	GGAATGAAGAAGCAAGCATCC	
T7-ds*SeCDA6*F	TAATACGACTCACTATAGGACACGAGTGCTTTGCTATTA	dsRNA synthesis
ds*SeCDA6*R	CCGTCGTAGCTGGAGATG	
ds*SeCDA6*F	ACACGAGTGCTTTGCTATTA	
T7-ds*SeCDA6*R	TAATACGACTCACTATAGGCCGTCGTAGCTGGAGATG	
T7-ds*SeCDA7*F	TAATACGACTCACTATAGGCGAAGAGTTGTGTAAGTTGCCT	
ds*SeCDA7*R	TGTTGCCAGCCAGTTGTAG	
ds*SeCDA7*F	CGAAGAGTTGTGTAAGTTGCCT	
T7-ds*SeCDA7*R	TAATACGACTCACTATAGGTGTTGCCAGCCAGTTGTAG	
T7-ds*SeCDA8*F	TAATACGACTCACTATAGGGCTACTACGCGCCTGACTTC	
ds*SeCDA8*R	GGAACTGAAATTGCACTGGTG	
ds*SeCDA8*F	GCTACTACGCGCCTGACTTC	
T7-ds*SeCDA8*R	TAATACGACTCACTATAGGGGAACTGAAATTGCACTGGTG	
T7-ds*SeCDA9*F	TAATACGACTCACTATAGGTGGCTGATTATGGTTTGGAG	
ds*SeCDA9*R	AGGATTCTTCACCCAGTCAA	
ds*SeCDA9*F	TGGCTGATTATGGTTTGGAG	
T7-ds*SeCDA9*R	TAATACGACTCACTATAGGAGGATTCTTCACCCAGTCAA	
T7-ds*SeCDAV*F	TAATACGACTCACTATAGGGTGAACGAGCTCTACAACCG	
ds*SeCDAV*R	GTTTAGGATGAACTGGAACC	
ds*SeCDAV*F	GTGAACGAGCTCTACAACCG	
T7-ds*SeCDAV*R	TAATACGACTCACTATAGGGTTTAGGATGAACTGGAACC	
T7-ds*GFP*F	TAATACGACTCACTATAGGCCACAAGTTCAGCGTGTCCG	
ds*GFP*R	AGTTCACCTTGATGCCGTTCT	
ds*GFP*F	CCACAAGTTCAGCGTGTCCG	
T7-ds*GFP*R	TAATACGACTCACTATAGGAGTTCACCTTGATGCCGTTCT	

## Data Availability

Not applicable.

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
