# Peer review of "Group V Chitin Deacetylases Influence the Structure and Composition of the Midgut of Beet Armyworm, Spodoptera exigua"

_ijms, 2023, doi:10.3390/ijms24043076_

Round 1
Reviewer 1 Report (Previous Reviewer 3)
The manuscript still need to futher improved and verify, I would like to give a acceptance as a stage development reseach.
Author Response
We thank the reviewer for reading our paper carefully and giving the above positive comments.
Reviewer 2 Report (Previous Reviewer 2)
The authors figured out most of the issues I concerned and made detailed responses to my comments. In view of the improved manuscript with much better quality, I suggest the publication of this work in this journal.
Author Response
We thank the reviewer for reading our paper carefully and giving the above positive comments.
Reviewer 3 Report (Previous Reviewer 1)
The English language used needs to be clearer. There are still some text errors when it comes to 'readability' such as:
Line 27 - "The SeCDA6/7/8/9s genes were down-regulated treated with 20-hydroxyecdysone (20E)."
- Is this after 20E treatment? This sentence doesn't make sense.
"After treating with juvenile hormone analog (JHA) treatment, SeCDA6 and SeCDA8 genes were expressed downregulateddecreaseing, while SeCDA7 and SeCDA9 genes were upregulatedincreasedon the opposite."
- spelling mistakes
- Use 'downregulated' and 'upregulated' to indicate changes in expression levels.
Line 57 - "Chitin is hydrolyzed by CHTs and NAGs, whereas CDAs deacetylasis is through an enzymatic process"
- Please define CHT and NAG in the text or spell out in the introduction as a baseline for the rest of the paper.
Line 91 - "Previous studies speculated on the function of Group V CDAs but did not discuss in-depth its role in the structure of the insect midgut"
Lots of other examples of English issues that detract from the overall quality of the manuscript
Author Response
Please see the attachment.

This manuscript is a resubmission of an earlier submission. The following is a list of the peer review reports and author responses from that submission.
Round 1
Reviewer 1 Report
Paper needs to be rejected on the grounds of no repeats for some experiments. Statistics performed on a single run of an experiment not possible.
Overall a nice paper, but needs to remove the pest control angle because no information about this is shown in any way.
Paper needs to be rejected on the grounds of no statistical test information provided and no original data available.
General notes:
1. Check typos and writing in general, not up to quality.
2. For figures with statistical tests, specify the tests performed and the number of individuals used to generate the data.
3. Update information for "Group V SeCDAs were silenced in larvae in- 160
jected with dsSeCDA6 and dsSeCDA8". This was hard to locate, specify in the figure legend and the materials and methods as well.
4. Writing needs to be improved for materials and methods significantly.
Line edits:
5 - 1 and 2 written for affiliations but should be superscript.
22 - "After silencing SeCDAV by interference (RNAi), the layer of mesenteric cells shrank and became more evenly distributed." What is SeCDAV? Is it a combinational knock down of all the other genes or is it another gene, there needs to be clariy here.
27 - "Obviously" Remove this word.
27 - "indicating that it is a potential target for S. exigua control." this statement doesn't make sense, if the insects didn't die then how is it useful for control, no mortality analyses are present so this sentence is irrelevant and should be removed.
64 - what is meant by 'nonviable phenotypes' clarify in writing here?
87-96 - writing is convoluted, just make a small table for the manuscript to include different information for each gene.
Figure 1 - no yellow or purple regions are seen so please update figure legend to reflect what's actually seen in the figure. Remove the CDA key in the figure since it's already in the legend and it squashes the aspect ratio of the figure.
Figure 2 - not useful, move to supplemental information.
110 - "done" Replace with 'performed'
113 - typo "distanceis"
163 - You have no real evidence of off target effects, it could be knockdown of one gene results on cascade based knockdown of another gene, could be that they are not direct off-target knock down but impacts of different RNAi trigger sequences or the interaction between dfferent regulation between these genes. Could be that there's expression co-dependencies going on.
167 - "After Group V SeCDAs were silenced, the larvae in control and treatment groups were normal and molted and pupated normally. 168" Where is the evidence for this? Provide it or say not shown.
Figure 9 - increase contrast for PI panel, stain barely visible. This is not a quantitative analysis so brightness & contract can be optimized for visualization but they need to be performed uniformly across the images.
297 - REMOVE BRACKETS THROUGHOUT E.G. (26.5±1) °C
320 - "fifth-instar larvae for total RNA extraction" how many and how many repeats?
345 - How many repeats?
Reviewer 2 Report
In the study titled with “Group V chitin deacetylase genes influence the structure and
composition of the midgut and peritrophic membrane of Spodoptera exigua (Lepidoptera)” by Han Wu et al. cloned and characterized the Group V Chitin deacetylase (CDA) enzymes in Spodoptera exigua. The author investigated the expression profiles of SeCDA 6-9 genes and explored the response after 20E injection. They also represent the morphological defects of mid-gut and peritrophic membrane after knockdown of SeCDA genes. The current results provided some information about the CDA enzyme in Spodoptera exigua which is similar to what has been found in other insect species. Some of the experiments and results are not quite solid or convincing to me and there are some weak points and concerns in the manuscript. An extensive modification and extra experiments are needed to solidify the overall conclusions made in the manuscript.
1. In the introduction section, the author introduced the research status of CDA gene in other insect species. Why did the author picked the group V CDA genes to perform further downstream studies? What are their criteria? The author needs to make a logic narration in the last paragraph of introduction section. As the author also did the phylogenetic analysis of group I-IV SeCDV genes in figure 3, they should add the information about the sequences and motif distributions about group I-IV SeCDV genes. These data or information could be put in the supplementary figures.
2. The expression data from figure 4 is kind of confusing to me which needs further modification. What group of data the author used for normalization which means it was set to 1? The author should re-organize the figure. For the expression of SeCDA7 and SeCDA9 , there is a huge decrease from embryonic stage to the first instar and a dramatic increase from first instar to the second instar larvae. Does the author has any reasonable explanation for that? Expect for the expression of SeCDA6 which is roughly stable during the larval stage, the expression of other three genes gradually increased and reached the peak at 5th instar or PP stage, does the author has any comments/narrations to explain the differences between them as they are all belong to the group V CDA? The author should add the related explanations or comments in the Results or Discussion section.
3. Please use the universal normalization method to re-organize the RT-PCR data presented in figure 5 and 6 which will make them more accessible to broader audience.
4. The author injected 20E to the animal and found the down-regulation of four SeSDA genes. What about the roles of JH, another important hormone in the insects development? What if the author inject the JH or JH analog to the animal, will we see the expression changes of CDA genes? I would like to see the this data presented.
5. The data presented in figure 8 does not make any sense to me. The author designed the gene specific dsRNA to knock down specific genes, while it seems that there is a sever off-target effect, especially for the dsSeCDA6 and dsSeCDA8. Even for the dsSeCDA9, it somehow induced the expression of othere SeCDA genes. The date presented means that these gene specific dsRNA is not useful at all. I think the author should get rid of the these data but keep the dsSeCDAV data. At the same time, the author should remove the deSeCDA7 images in figure 10 and 11, then combine the figure 10 and figure 11.
6. The author should re-do the immunostaining in figure 9 and include the DAPI which will make it easier to see the thinner mesenteric cell layer. At the same time, with DAPI signal, the author could also check whether the thinner mesenteric cell layer is due to the reduce of cell size or the decrease of cell number. This might provide us some information about the potential functions of SeCDA genes in mesenteric cell survival or maintenance.
Minor points:
1.Line 132-133, “As midgut-specific proteins SeCDA6, SeCDA7, SeCDA8, and SeCDA9 132 of Group V are highly expressed in the midgut (Figure 5).” How does the author these proteins are “mid-gut specific” before check their expressions? The author should re-write this sentence.
Reviewer 3 Report
main comments were as follows:
1、There’re too many figures, should merge some figures and preferably not more than 8.
2、“The findings of this study can be useful in the selection of environmentally friendly RNAi-mediated insecticidal targets”, so how about the mortality and probable control effect? There should be more visualized phenotypes.
3、Totally 5 genes, how about the relationship among these genes? Which one is the key gene?
4、“after 20-hydroxyecdysone (20E) treatment, SeCDA6/7/8/9 genes were down-regulated, indicating that 20E regulates CDA genes”, there should be more verification and explain the relationship between 20E and CDA genes.
Round 2
Reviewer 2 Report
1. The overall organization of the manuscript is somehow disrupted because of the formatting, especially the figures, please modify it and make it easier to review.
2.Where are the references which support the author’s comments in line 79-81. Please add related references to the context.
3. Where is the supplementary fig.1? I did not see any supplementary information files. Please attached these information for review.
4. I did not see the normalization of data in figure3. If the author used the pupae stage data for normalization, why the pupae stage data was not set to 1?
5.There are duplicate figures in current figure3.
6.The author ignored my comments about the universal normalization and just add few words in the figure legend. I did not see any changes in figures from the revised the manuscript!
7. I need to wait and see the author’s JH injection results. Please show me these data after your experiments in revised version.
8.There are duplicate figures in current figure 7.